# Multi-Steps Magnetic Flux Entrance/Exit at Thermomagnetic Avalanches in the Plates of Hard Superconductors

**DOI:** 10.3390/ma15062037

**Published:** 2022-03-10

**Authors:** Viktor Chabanenko, Adam Nabiałek, Roman Puźniak

**Affiliations:** 1O. Galkin Donetsk Institute for Physics and Engineering, National Academy of Science, Pr. Nauki 46, 03028 Kyiv, Ukraine; 2Institute of Physics, Polish Academy of Sciences, Aleja Lotników 32/46, PL-02668 Warsaw, Poland; nabia@ifpan.edu.pl (A.N.); puzni@ifpan.edu.pl (R.P.)

**Keywords:** hard superconductors type II, thermomagnetic instability, avalanche flux dynamics, screening and trapping regimes, cascades of flux

## Abstract

Avalanche cascades of magnetic flux have been detected at thermomagnetic instability of the critical state in the plates of Nb-Ti alloy. It was found that, the magnetic flux Φ enters conventional superconductor in screening regime and leaves in trapping regime in the form of a multistage “stairways”, with the structure dependent on the magnetic field strength and magnetic history, with approximately equal successive portions ΔΦ in temporal Φ(*t*) dependence, and with the width depending almost linearly on the plate thickness. The steady generation of cascades was observed for the full remagnetization cycle in the field of 2–4 T. The structure of inductive signal becomes complex already in the field of 0–2 T and it was shown, on the base of Fourier analysis, that, the avalanche flux dynamic produces, in this field range, multiple harmonics of the electric field. The physical reason of complex spectrum of the low-field avalanche dynamics can be associated with rough structure of moving flux front and with inhomogeneous relief of induction. It was established that the initiation of cascades occurs mainly in the central part of the lateral surface. The mechanism of cascades generation seems to be connected to the “resonator’s properties” of the plates.

## 1. Introduction

Magnetic properties of hard type-II superconductors are usually described in the frames of Bean critical state model [1,2]. In this model, the superconductor volume is screened by a current of the critical density, *j*_c_. In increasing external magnetic field, magnetic flux enters into the superconductor’s volume as a series of tiny flux jumps of different scales [3]. These jumps allow to relax the critical state and were studied experimentally [4,5].

In the terms of thermodynamics, the critical state is metastable. At certain conditions, small fluctuation of an external magnetic field, temperature, or tiny magnetization fluctuations in superconductor may lead to appearance of a catastrophic thermo-magnetic avalanche [6,7]. Magnetic flux jumps and accompanying heat release are the phenomena commonly observed both in conventional and in high temperature superconductors [8]. During the thermo-magnetic avalanche, the magnetic flux enters/exits into/from superconductors abruptly and magnetic moment decreases sharply. The temperature range, in which the critical state is unstable, is determined by the material parameters and can reach the temperatures as high as ten Kelvin. Low temperatures, where the critical current density increases significantly, are especially interesting for practical application. However, in this case, the instability of the critical state increases strongly too, leading to a giant flux avalanches.

An experimental study of avalanche flux dynamics, in particular, visualization of magnetic field penetration patterns, began more than fifty years ago [9,10,11,12,13]. Direct observation with high speed cinematography of the flux distribution and its change in time over the entire surface of a superconducting sample made it possible to study the kinetic of flux jumps. Particular attention was paid to the study of the flux front velocity, both in pure metals and in alloys [10,11,12,13,14]. It was found that the flux front velocity for conventional superconductors is of an order of tens of meters per second [11,14]. At the same time, the structure of magnetic flux avalanche spot in disks was found to be very diverse: from circular spots [12] to flux fingers and “dendrite-like” structures (“irregular” jumps [13]) in inhomogeneous bulk superconductors. Here, “irregular” jumps are the jumps leading to a complex, irregular distribution of the magnetic flux in the superconductor. The analysis of the visualization patterns of the flux dynamics, presented in the above papers, allowed, within the framework of Bean’s concept, to understand the mechanism of local inversion of the surface self-field in the volume of conventional NbTi superconductor [15,16] and the complete inversion of the magnetic moment in the YBa_2_Cu_3_O_7−δ_ single crystal [17].

Avalanche flux dynamics in thin films, with penetration field depth greater than their thickness, attracts special attention because of several phenomena, recently discovered by means of magneto-optic, including dendritic flux patterns [18,19,20,21,22,23,24]. Such avalanches consist of sharp impulses of the magnetic flux rushing into the specimen. The “branches” of dendritic structures repel each other as they grow. Numerical simulations of the vortex avalanches in superconducting films confirmed a phenomenon of magnetic flux fragmentation [25,26], occurring when the hot spot of the thermomagnetic avalanche reflects from inhomogeneities or the boundary, on which magnetization currents either vanish or change direction. As a result, thermomagnetic avalanche produce complex flux patterns similar to dendrite.

Variation of Nb film geometry extended diversity of magnetic flux structures that penetrate superconductor during avalanches. One of such structures is referred to as “huge compact avalanches” [27] and it is different from dendrites. A recent study found an unambiguous relationship between the size and shape distributions of avalanches and thermomagnetic conditions of instability development [28]. It was established that the transition from a curved “finger-like” structure of avalanche at low fields to a dendritic one at higher fields allows dividing the regimes of dynamical process, where either thermal or magnetic diffusivity prevail.

In addition to the variety of avalanche structures, the study of the dynamic properties of the flux led to an unexpected result: the dendrite front propagation reaches the velocity up to 160 km/s [18,22]. An impressive experimental fact, discovered recently, is the rate of nanoportions of the flux, the Abrikosov vortex, penetrating into the superconducting Pb film [29,30] or into the Nb-C superconductor [31], being up to tens of kilometers per second and exceeding the theoretical estimates of pair breaking speed limit for superconducting condensate. Moreover, as it was shown, calculating the electromagnetic response of a nonequilibrium superconducting state, much lower velocities of the superconducting condensate lead to the appearance of multiple harmonics of the electric field [32], which means that the condensate dynamics is a highly nonlinear process. These results indicate that the study of the dynamic properties of magnetic flux helps to uncover unknown properties of the superconducting state. However, limited attention only was paid to the effective mass of Abrikosov’s vortex [33,34,35], which can manifest itself, for example, in the experiments with circular dichroism of far-infrared light [36].

The purpose of current research was to establish the regularities of dynamic processes, occurring in a superconductor, when the path length of the finger-shaped avalanche is significantly limited by the size of the superconductor, i.e., it was supposed to implement the conditions of the “resonator” for a directed electromagnetic shot in the plates of the most popular hard superconductor. We present, here, an evidence of a multi-step “stairway” structure of magnetic flux dynamics at the thermomagnetic avalanches in the volume of the plates of conventional NbTi superconductor, being the number one material in the world in the terms of wide application. The width of the step, depending on the thickness of the plate, was determined in inductive measurements. The phenomenon of multi-steps magnetic flux dynamics was observed, both at the flux avalanche entry (screening mode) and at its exit (trapping mode). Spectral analysis of the structure of inductive signal allowed us to characterize the avalanche flux dynamics in the entire region of instability of the critical state, and to establish the region of magnetic fields where steady state generation of cascades is realized. The experimental observation of multi-step “stairway” structure of magnetic flux dynamic at avalanches is reported here for the first time.

The experimental results presented here demonstrate the new dynamical properties, appearing during thermomagnetic avalanches, in the plates of type II superconductors. It was found that, the magnetic flux enters conventional superconductor in screening regime and leaves in trapping regime in the form of a multistage “stairways” (cascades), with the structure dependent on the magnetic field strength and magnetic history, with approximately equal successive portions in temporal dependence, and with the width depending almost linearly on the plate thickness. The mechanism of cascades generation seems to be connected to the “resonator’s properties” of the plates.

## 2. Experiments and Materials

Time resolved measurements were performed with the aid of different type of inductive sensors, where a voltage (*U*_coil_ (*t*)~dΦ/d*t*, *t*—time) caused by the avalanche flux jump was induced. Figure 1a shows flux jump at *T* = 1.7 K in Nb disc, frozen in the form of “fingers”. Figure 1b–d shows the arrangement of pickup coils on superconducting plate for the avalanche registration: in the whole plate’s volume (b), on the three height levels (c), and in the magnetic stray field (*L*_ext_) (d). Time-dependent voltage *U*_coil_ (*t*) was registered with NI DAQ 6115S (National Instruments, Austin, TX, USA) (data acquisition board) simultaneously in four channels with time resolution of ~10^−7^ s. External magnetic field (*H*_ext_) applied in direction parallel to the plate was swept with the rate of 0.6 T/min and local magnetic induction (*B*_surf_) at the center of the sample surface was measured with Hall sensor. The magnetization *M* of superconducting plate was determined by the difference between local and external magnetic field induction, µ_0_*M* = *B*_surf_ − µ_0_*H*_ext_. The specimen was immersed in liquid helium bath and the temperature down to 2 K was reached in pumped helium.

Initial plate with the size of 20 × 14 × 7 mm^3^ has been cut from extruded cylindrical rod of NbTi 50 at% alloy with diameter of 15 mm. Hot extrusion of Nb-Ti 50% alloy was carried out according to the standard technology [37] through a deformation of matrix at a temperature of 750 °C along the route from Ø 50 mm → Ø 15 mm (Ø—diameter of the rod) with a draw value *R* = *S*_before_/*S*_after_ ≈ 11, where *S*_before_ and *S*_after_ are the areas of the sample sections before and after deformation, respectively. The side surfaces of the plate were ground with diamond (corundum) powder in order to remove the layers of the material strongly deformed during cutting. Such procedure was repeated at each stepwise thinning of the plate to the required thickness *d*. The pickup coils were close-wound with 0.1–0.2 mm diameter of copper wire and consisted of 15–100 turns.

## 3. Results and Discussion

The salient features of the manifestation of thermomagnetic avalanches in the plates at 2 K are their realization in the form of a multistage “stairway” manner of magnetic flux dynamics (Figure 2a,b). One example of the voltage *U*_coil_(*t*) signal inherent to a cascade induced on a pick up coil at the avalanche in trapping regime is presented in Figure 2a (2nd quadrant of hysteresis loop on Figure 2e, Jump 34). The signal consists of a series of regularly spaced impulses (portioned avalanches) and shows a multi-step “stairway” structure of magnetic flux dynamic [Φ(*t*)], received as a result of integration of voltage *U*_coil_(*t*) curve (Figure 2b below Figure 2a). Thus, the magnetic flux enters (in screening regime) or leaves superconductor (in trapping regime) in “steps-like” manner (Figure 2b). Such well-structured cascades are observed at the temperature of 2 K for plates of different thicknesses with *d* = 2.7–6 mm in a certain region of the magnetic field, marked with rectangles and arrows on the hysteresis loops *M*(*H*_ext_) (Figure 2d–f, right column). Remagnetization loop at 4.2 K for the plate with thickness of 6 mm—for comparison with *M*(*H*_ext_) curve at 2 K—is shown in Figure 2c.

Figure 3 presents a panorama of the voltage impulses structures *U*_coil_(*t*) at 2 K, induced on a pick up coil at the flux cascades, originating for the plates with various thickness, i.e., with thickness of 2.7 mm (a), 3.1 mm (b), 4 mm (c), and 6 mm (d). Left side of each figure contains cascades in screening mode (1st quadrant) and the right one in trapped mode (2nd quadrant). The most regular step-like structures were found for the plates with diameter of 2.7 mm, 3.1 mm, and 4 mm. Nevertheless, the step-like structure is also visible in a narrow range of magnetic fields for the plate with a diameter of 6 mm (Figure 3d). The maximal number of steps in the cascades was found for magnetic field in the range of 2–4 T.

### 3.1. Evolution of the Structure of Cascades at the Change of Magnetic Field

*Screening regime.* Time evolution of the voltage impulses *U*_coil_(*t*) at *T* = 2 K at the avalanches with increasing magnetic field in screening regime, measured after cooling in zero magnetic field (ZFC—zero field cooling), for the 1st quadrant in the plate with *d* = 2.7 mm, is shown in Figure 4a. Amplitude of the signals presented was normalized to the maximal peak value *U*^max^_coil_ for each cascade. The absolute value of voltage impulses decreases exponentially with increasing external magnetic field, as it is presented in Figure 4b. The structure of the avalanche is relatively simple for the first and for the last jump: only one single peak of voltage was found, as it is seen in Figure 4a.

However, for the first avalanche already, the process of excitation or triggering of cascades of avalanches is observed (Figure 5a), despite that, the “oscillations” *U*(*t*) associated with this process are still weakly expressed. With increasing magnetic field, the voltage impulse expands into a shape of a crown with numerous peaks at the fourth avalanche (Figure 5b), and the step structure of the flux becomes apparent. In stronger fields, the “crown” is transformed into a structure of clearly separated, isolated avalanche impulses (Figure 3a and Figure 4a). It means that, the whole avalanche, occurring in the period of several tens of milliseconds, exhibits a multi-step structure, i.e., the magnetic flux enters (leaves) in approximately equal, successive portions ΔΦ, the number of which can reach the value of ten (Figure 2b and Figure 3). The range of magnetic field between 2 and 4 T may be defined as a range of steady state of cascades generation. It should be noted that duration of entering of each individual portion of the flow ΔΦ in the cascade is the same as in the single avalanches that are observed at the beginning and at the end of the region of instability of the critical state (Figure 4a). The nucleation of cascades is shifted into stronger magnetic fields with thickening the plate to *d* = 3.1 mm (Figure 5c).

It should be noted that with an increase in temperature to 4.2 K, the signal from the excitations of the cascades is practically not observed on the avalanche pulse (Figure 4c, *d* = 4 mm). At the same time, one can see, on the trailing part of the signal, a certain sequence of small peaks of the flux entry.

### 3.2. The Flux Step-Like Structure in Avalanche Cascades

*Periodicity analysis.* To study the patterns of stepped structure of the flux Φ(*t*) (Figure 2b) under the influence of external factors, its derivative *U*_coil_(*t*) should be analyzed, which allows to consider changes in the repetition period *T* of individual avalanche pulses (flux portions) in a cascade, or to analyze their characteristic frequencies *F* (pulses) of their repetition. For simple harmonic signals, this is, in principle the same, since the relationship between these quantities is trivial: *T* = 1/*F*. In the case of the studies performed here, it is not always easy to visually determine the period from the signal structure. This is especially true for the region of relatively weak magnetic fields −1.5 T–0–+1.5 T in the second and third quadrants.

Let us first consider the periods *T*_c_ between peaks of voltage *U*_coil_(*t*) in cascades for different forms of studied signal. In Figure 6a,c and Figure 7a,c, some characteristic and their detailed analysis, for 2.7 mm and 3.1 mm samples, respectively, are shown, with the voltage impulse *U*_coil_(*t*) (left ordinate) inherent to a cascade at the avalanche, and a multi-step “stairway” structure of magnetic flux Φ(*t*) (right ordinate) in a screening (a) and a trapping (c) regime. Two types of signals have been selected here: in the first case (a) avalanche impulses overlap slightly (the end of the “crown”, Figure 4a), in the second—Figure 6c—avalanche impulse are spaced (separated) in time. The latter case is typical for avalanches in strong magnetic field, near the boundaries of the region of instability of the critical state.

The periods *T*_c_ between jumps in cascade vs number of impulses *n* (or steps) are shown in the main frame of Figure 6b,d, left ordinate. As follows from presented data, the screening and trapping regimes *T*_c_ increase over time in both cases. The experimental points are scattered around the straight lines, constructed with least squares method. The values of the frequency pulse *F*_T_ = 1/*T*_av_ in cascades (*T*_av_ is the average period) are given below the lines. Importantly, a similar period is observed in three quadrants for the plates of different thicknesses, in the range of steady state of cascades generation.

The increase in the *T*_c_ period with time can be associated with an increase in the dissipation of the induction-current system in the critical state as the flux enters the superconductor. Such increase is characteristic for oscillatory with an increase in the attenuation. Structured cascades with a uniquely visually identified frequency rarely appear during magnetization reversal in the range of 0–±1.5 T, in the second and third quadrants. Here, avalanches exhibit, mainly, a complex structure. An example of the signals for a plate with a thickness of *d* = 3.1 mm, where it is difficult to visually assess the main period of *T*_c_ and the corresponding frequency, is presented in Figure 7c. Here, spectral analysis helps to establish the frequency components of the spectrum and, if necessary, to determine the appropriate periods.

*Spectral analysis.* With an increase in the magnetic field, the structure of avalanche pulses passes smoothly from a single pulse with small oscillations (Figure 5a) to almost periodic oscillations limited in time (Figure 3). Proper analysis of the data requires information on the amplitude and frequency of the signal spectrum components, i.e., on the amplitude-frequency spectrum and thus, fast Fourier transformation (FFT) from the Origin program (OriginLab Corporation, Northampton, MA, USA) was utilized in the analysis of the *U*_coil_(*t*) spectra in the entire region of instability of the critical state. The characteristic *F*_FFT_ frequencies in the signal spectrum, determined with this package and compared with the values of the pulse repetition frequency *F*_T_, obtained directly from the average period *T*_av_, exhibited its good efficiency (see, the inset in Figure 6b,d and Figure 7b,d—the amplitude spectra of avalanche signals with characteristic *F*_FFT_ frequencies). The values of the pulse frequency gained from the average period *F*_T_ for plates with a thickness of *d* = 2.7 mm and 3.1 mm, are presented in Figure 6 and Figure 7, respectively. The spectrum component with the maximum amplitude presented in Figure 6b corresponds to the frequency *F*_ma_ = 952 Hz, which is in good agreement with the frequency value estimated from the average period: *F*_T_ = 927 Hz. The spectrum shown in the inset in Figure 6d is more complex. Here, there is a fundamental frequency *F*_ma_ = 470 Hz with a maximum amplitude, which practically coincides with the frequency *F*_T_ = 487 Hz. In addition, the spectrum contains two more components, the amplitude of which exceeds the amplitude level of 50% of the fundamental harmonic. A good agreement of spectral analysis is presented in Figure 7: *F*_ma_ = 724 Hz and *F*_T_ = 730 Hz (b) and *F*_ma_ = 551 Hz and *F*_T_ = 564 Hz (d).

It should be noted that the signal *U*_coil_(*t*) presented in Figure 7c exhibits a spectrum with three component frequencies (Figure 7d), with similar amplitude values. This is definitely due to the strong beats in the signal. Apparently, the presence of such frequency components, with close amplitude values, leads to “jumps” in the experimental dependences of the *T*_c_ periods vs number *n* (Figure 6b,d) around the approximating straight lines. The data indicate that the harmonic *F*_ma_, with the maximum amplitude in the signal spectrum, is suitable for analyzing the effect of external influences and parameters of a superconducting plate on the phenomenon of multi-steps magnetic flux dynamics in the entire region of instability of the critical state.

*The results of studying signals from avalanches* for three quadrants in a plate with a thickness of 3.1 mm are shown in Figure 8 and the features of the hysteresis loop *M*(*H*_ext_) associated with the excitation of cascades are shown in Figure 8a. A stepwise decrease in the amplitudes of magnetization jumps Δ*M* is clearly expressed in the region of cascades for three quadrants of hysteresis loop and for all thicknesses (Figure 2d,e), which is a result of the excitation of cascades. This is evidenced by magnetic field dependence of the avalanche duration Δ*t*(*H*), shown in Figure 8b (main frame) and by their structures in the region of the amplitude step in field dependence of magnetization. The avalanches in this place [inserts into (b), jump 5 and 6] undergo threshold changes. During the sixth avalanche, the origin of multistage nature of the flux dynamics sharply increases, while the duration of the avalanche suddenly increases by three times, and the incoming flux is divided into six portions spaced in time (jump no. 6). The explanation of the step in magnetization is quite simple. Step-like behavior is associated with a change in the localization of the avalanche spot of the incoming flux and with Hall sensor method of induction measurements, where a local field change at the center of the surface is registered. During avalanche no. 5, the magnetic spot was located mainly in the central part, under the sensor. In the case of cascades (avalanche 6), the magnetic flux was dispersed like a comb of six teeth along the entire side of the plate. In this case, the level of induction under the sensor becomes significantly smaller.

Here, it is appropriate to emphasize practical implication of the importance of phenomenon of cascades at avalanche in the volume of superconductors: the incoming avalanche flux is dispersed in time and becomes non-localized in high field region. One of the main problem challenges in superconducting applications is to avoid thermomagnetic breakdown of critical state (quench) and as a result, the sudden disappearance of beneficial properties. To avoid creation of conditions for non-localized place of quench is one of the important practical issues solved by technologists today [38]. The discovery of a multi-steps magnetic flux entrance/exit at thermomagnetic avalanches in the plates of hard superconductors could make a difference in the case of applications of electric motors or generators built with superconducting elements in the form of plates. In the plates with certain thicknesses, this phenomenon reduces the risk of failure at maximum loads.

The evolution of the avalanche structure in the first quadrant (shielding mode) was previously analyzed in detail, using the example of a plate with *d* = 2.7 mm (Figure 4a and Figure 5). Now, we will consider the properties of cascades in the second and third quadrants. The instability of the critical state is observed here in a field range from +4T to −4T (Figure 8a,c). The frequency values *F*_ma_ for the components with the maximum amplitude in the signal spectra are shown in Figure 8c (main frame). The *F*_ma_ frequencies, in the range of high magnetic field ±2–±4 T (steady state cascades generation), are marked by full squares (■). Here, the frequency decreases linearly with an increase of magnetic field for all quadrants. Such behavior of the *F*_ma_ may be interpreted as a result of increasing of dynamical dissipation of critical state and by decreasing of the avalanche start velocity as a result of the critical current density *j*_c_(*H*_ext_) decreasing.

The stable cascades with a small number of flux steps ΔΦ appear in the field region −2 T–+2 T, as it is shown, for example, in the left and in the center inserts at the top of Figure 8c. The data in the main panel, described by asterisks (☆), indicate the frequencies *F*_ma_. The signal *U*_coil_(*t*) exhibits here a complex structure with a wide spectrum (Figure 7c,d) and it seems that, in such a case, determination of the fundamental harmonic *F*_ma_ (asterisks in Figure 8b) is very difficult, if it is possible at all. Therefore, an attempt was made to present, for each signal *U*_coil_(*t*), the frequencies of all spectrum components with amplitudes higher than 50% of the maximal value. The result for the second quadrant is shown in Figure 8d. Here, the frequencies *F*_ma_, described by asterisks (☆), as in Figure 8c, are shown, together with frequencies of the harmonics with slightly lower amplitudes, described by the full circles (●). In the field region *H* 2 T, one can visually distinguish between certain lines, designated with the numbers 1, 2, and 3, on which most of the experimental points lie on the array. Field dependence for lines 1 and 3 is similar to that of line 2, which is a continuation of the straight line *F*_c_(*H*_ext_) from the field range of steady state generation of cascades. In terminology taken from other branches of physics, the lines 1, 2, and 3 can be described as follows: Frequency “level” 2 (line 2) for *F*_ma_(*H*_ext_) corresponds to the dynamic state of the flux with steady state generation of cascades and it splits in a magnetic fields below 2 T into two “levels”, corresponding to the states of lines 1 and 3. Such behavior can be expressed as a certain degeneration of the “frequency levels” on lines 1 and 3, which occurs in a magnetic field µ_0_*H*_ext_ 2 T.

We managed to observe, for one of the cascades in the transition region (inset in Figure 8d), a signal, occurring at the boundary in *H*_ext_ = 2.01 T, where two frequencies marked by crossed open circles (⊕) are clearly visually present: the upper *F* = 1040 Hz belongs to the straight line 1 and the lower frequency *F* = 674 Hz, which falls on line 2 for frequencies corresponding to stable cascades. Analysis of the avalanche flux dynamics in the second quadrant indicates the presence, in the region of weak fields (µ_0_*H*_ext_ 2 T), of some effective mechanism, leading to the appearance of electromagnetic radiation of a complex spectral composition. In a thinner plate with *d* = 2.7 mm, the field dependences of the frequency *F*_ma_, which characterize the avalanche flux dynamics look similar—Figure 9a. In the range of steady state generation of cascades, the frequency dependences *F*_ma_(*H*_ext_) for the first and third quadrants practically coincide. The data for the second quadrant also exhibit a significant “scatter-rocking” of the experimental values of *F*_ma_(*H*_ext_) and are lowered, in the same field range, in comparison with those in the first quadrant. Two measurements are presented for the second quadrant [marked by two different kind of stars (☆)] in order to show the repeatability and rocking range of experimental data in complex spectra.

The amount of magnetic fluxes in avalanches significantly differs between the first and the second quadrant (Figure 9b): in the same magnetic field much more flux enters in the shielding mode than exits in the trapping mode. Moreover, in the latter case, experimental values for the exiting fluxes are extensively scattered, as do the values of the frequencies in the dependence *F*_ma_(*H*_ext_). It should be noted that the amount of magnetic fluxes decreases exponentially with increasing external magnetic field for the 1st quadrant (Figure 9b) as well as the absolute values of voltage impulse (Figure 4b). Figure 9c shows the period *T*_ma_ = 1/*F*_ma_ between jumps in cascade as a function of magnetic field for various quadrants. It becomes clear here, the pulse frequency in the inductive signal depends linearly on magnetic field.

*The role of plate thickness.* Magnetic field dependence of the jump frequency *F*_ma_ for cascades in screening regime (1st quadrant) for the plates with various thickness *d* is shown in Figure 10. It can be seen from these data that the region of the magnetic field, where cascades are stably realized, rises up along the frequency axis when the plate becomes thinner. At the same time, the range of magnetic fields and frequencies of cascades is significantly expanded. Insert in Figure 10 shows the dependence of period 1/*F*_ma_ between jumps in cascades as a function of plate thickness along vertical lines at µ_0_*H*_ext_ = 2.5 and 3 T in the main frame. The period between avalanches inside cascades increases almost linearly with increasing plate thickness. It may indicate on the functional role of the plate thickness in triggering of cascades.

### 3.3. Triggering, Propagation and Localization of the Avalanche Flux Cascades

*The place of avalanche origin.* An avalanche appears, usually, in a small volume, almost at a point compared to the size of the plate, and a bundle of vortices trapped at pinning centers can stimulate an avalanche. The process of magnetic field penetration constantly accompanies jumping of the bundles [3,4]. Simultaneous registration of the signal at different levels along the height of the plate (Figure 1c) opened possibility to find the level where the signal appears first and to estimate the propagation velocity of the disturbance along the magnetic induction line. The analysis of the time delay Δ*t*_delay_, between signals (more than 60 signals) from the avalanche beginning, allowed us to fix the most probable place of their origin. Statistics showed that the avalanches originate in 68% near the central region (CP, Figure 1c) of the lateral surface of the plate, and in 32% only—at its edges. This result agrees with calculation of field penetration into the plate [39]. In the central part, the screening is maximal, the magnetic field pressure is the highest, and, accordingly, the initiation of instability is most likely here. An attempt, to find side surface, where an avalanche originates (and to feel from which side a separate jump of the flux enters), using two external inductive sensors on different sides of the plate (*L*_ext_, Figure 1d), while recording signals, was unsuccessful. It means that, the delay (or accelerates) of the avalanche onset on one of the lateral surfaces and the asymmetry of scattering field dynamics on opposite sides of the plate, during the stepwise entry of the avalanches, were not detectable. A beautiful shaped cascade when the signals from two sensors were added and its almost complete compensation to the noise level, when they were subtracted, was observed. It should be noted, here, that, in wide coils, covering a significant part of the side surface of the plate, the stages looked to be more structured. This may be due to the fact that the origin of the cascades travels along the vertical or along the lateral surface and their appearance is somewhat different in different sections. Most probable speed of propagation of the disturbance along the direction of the magnetic field, *V*_||_, was determined on the basis of statistical data analysis and it was found that, it corresponds to the maximum velocity distribution function. The velocity reaches *V*_||max_ = 0.5 *h*/Δ*t*_delay_ = 286 ± 20 m/s, where *h* is the plate height.

*Spatial-temporal cascades behavior*. In order to establish the location of the parts of the incoming flux on the central section of the plate, the pickup coils for the avalanche flux registration were placed on superconducting plate (Figure 11a) in the whole plate’s width (*W*) − (*L*), in the 1/3*W* (*L*_1/3_) and 2/3*W* (*L*_2/3_). In order to cover individual sections of the plate, a hole with a diameter of 0.7 mm was made for the placement of inductive sensors *L*_1/3_ and *L*_2/3_. Self-consistent cascade for the plate with *d* = 3.1 mm, recorded at *T* = 2 K in µ_0_*H*_ext_ = 3.35 T simultaneously by three coils *L*, *L*_1/3_, and *L*_2/3_, is shown in Figure 11b. Full signal registered by coil *L* is divided proportionally between the coils *L*_1/3_ and *L*_2/3_, confirming spatial separation of impulses in the cascades along the lateral surface of the central section. Schematic view of possible spatial localization of successive flux avalanches (steps) in the form of fingers is shown on the cross-section of a plate orthogonally to magnetic field (Figure 11c). The numbers given in Figure 11b,c indicate the sequence of avalanches occurrence. The individual flux pulses are shown here in the form of “fingers” since it is known [13] that the shape of avalanche spots transforms at low temperatures from round at *T* = 4.2 K to “finger-like” at 1.8 K. “Fingers” of avalanches in the scheme (Figure 11c) cross the plate all the way to the opposite side, rather than stopping at its middle, which is in agreement with the results of experimental studies of the distribution of surface induction—signal registered by an array of the Hall probes (Figure 1b) before and after appearance of thermomagnetic avalanches in increasing and decreasing external magnetic field [15]. Here, local induction inversion as a result of a thermomagnetic avalanche over the entire thickness (*d* = 4 mm) of the studied NbTi plate occurred already at *T* = 4.2 K (Figure 3b,c from Ref. [15]), which may indicate that the avalanche front crosses the entire plate. However, an unambiguous answer can only be given by direct observation of the flux front using magneto-optics.

Simulation of avalanches movement in superconducting films [25,26] indicates that the avalanche front tends to displace the superconducting current from the surface to the opposite side of the sample. It means that the superconducting current is pushed out of the hot zone of the avalanche finger into the cold zone of the superconductor [40]. In shielding mode (diamagnetic induction), an appearance of “paramagnetic” profile is expected if the flux front manages to cross the middle of the sample, which takes place in the simulation. This is exactly what was observed in the experiment (Figure 3b from Ref. [15]).

### 3.4. Possible Mechanism of Cascades Triggering

The linear dependence of the cascades period on plate thickness suggests that the period of step structure of magnetic flux avalanche dynamics Φ(*t*) in cascades may be controlled by plate thickness as well as the limitation of path length of the front of an individual avalanche finger in the cascade, introduced by plate thickness, can lead to the appearance of a certain sequence of excitation in the induction-current system of the critical state of the superconductor.

Let us consider a possible scenario of cascades triggering. When the critical induction drop Δ*Β*_FJ_ is reached, the initial flux step (finger) in cascades can be stimulated in the superconductor critical state layer by small jumps of the flux bundles. Penetrated first finger’s flux redistributes current in the sample, and thus creates weak spots for penetration of the consequent avalanche finger, etc., [40]. Weak spots appear in the concave corner (Figure 26 from Ref. [41]), near the finger’s gate (Figure 11c), where the current lines looming (become thicken) occur. Subsequent avalanches in the cascade can trigger a weak spot by a locally amplified pulse of a magnetic or electric field [42] and, accordingly, a current Δ*J* due to deceleration of the previous finger of the flux reaches the opposite side of the plate. The plate in this way can show its “resonator” properties.

The expansion of the range of magnetic fields, where cascades stably arise, with decreasing plate thickness, as well as the disappearance of this phenomenon for plates with *d* 6 mm (Figure 10), testifies in favor of the important role of finger’s-type flux reflection from the opposite side (resonator properties of the plate) in the process of self-excitation. 

Let us try to estimate the time it takes for the flux front to cross our plate in order to compare it with the period of multistage “stairway” structures. The dynamics of the avalanche front in conventional superconductors Nb and NbZr at a temperature of *T* = 1.4 K and in a magnetic field of 0.2 T was studied by the magneto-optical method [14] and time dependence of path traveled by the avalanche front is shown here. The magnetic flux front runs a distance of 3 mm in 0.4 ms for NbZr and in 0.6 ms for Nb, respectively, which is plotted in Figure 9c, presenting the data on the pulse periodicity in cascades for a 2.7-mm thick plate, corresponding well to the values of the periods of cascades. Hence, we can conclude that the plate, showing its “resonator” properties, can serve as a metronome that sets the period of magnetic flux dynamics in plates and lead to a multi-step “stairway” structure of flux Φ(*t*). Existence of avalanches in the form of a cascade of almost equidistant events can be unambiguously confirmed by simulating it only. However, it is quite plausible since penetrated flux redistributes current in the sample, and thus creates weak spots for penetration of the consequent avalanches, etc., [40].

### 3.5. Impact of Magnetic Prehistory

The change in the profile of magnetic field in the plate, with the change from increasing to decreasing field in the second quadrant, leads to a displacement of the boundary of critical state *H*^2^_tfj_ stability region into weaker fields (Figure 8b), as compared to the first quadrant (*H*^1^_tfj_). The physical reason for this is a decrease in the critical current density due to the trapped flux, since the trapped flux increases, in a given external magnetic field, the average effective field in superconductor, as compared to that in shielding regime [43]. The profile of magnetic field induction can affect the spectral characteristics of the avalanche dynamics of the front flux. The magnitude of induction gradient in superconductor sets the impulse to the flux in the avalanche at the “start”. The subsequent movement of the avalanche front in the plate passes along the induction relief formed before this, with the exception of the first avalanche in the screening mode. A schematic representation of Bean’s magnetic induction profile *B*(*x*) in a plate corresponding to situation before first flux jumps in screening regime and for trapping regime in strong magnetic field (field ramping backward) is shown in Figure 12 [left side of (a) and (b), respectively]. The result of avalanche flux entry/exits process is presented on the right side of Figure 12a,b. Here, a cross-section of a plate perpendicular to magnetic field is shown.

*Screening regime.* Avalanches appear above the field of the first instability *H*_ext_
*H*_1fj_ (Figure 12a) and the regime of cascades generation appears at the first avalanche, as it was depicted in Figure 5a. The critical current is maximal here, the pressure is high, and the flux front speed is high. Thus, a large amount of magnetic flux enters the region of the Meissner state of the superconductor (Figure 12a, right side). The central strong avalanche appeared, apparently, in the form of a finger. However, when its propagation was limited by the second side of the plate, the width of the finger under the pressure of the flow of vortices began to increase in the orthogonal direction, occupying a significant part of the plate. According to the signal structure (Figure 5a), subsequent flux fingers in the cascade were weaker. As the magnetic field increases, the critical current density decreases. Therefore, in the next cascades, the amount of flux that entered at the first pulse decreases, and to fill the plate with the flux, the number of subsequent avalanches in the cascade increases.

*Trapping regime.* The induction-current structure of the critical state in the flux trapping mode is fundamentally different from that in the screening mode. With shielding, only one critical current direction is present (Figure 12a, right side). In the second quadrant, due to a change from increase to decrease of external field, a second current loop in the opposite direction appears in the current structure (Figure 12b, right side). In this case, a boundary arises that separates these counter currents. In Bean model it is a straight line. In reality, due to the inhomogeneity of the pinning in the superconductor or the stochasticity of the vortex dynamics, this boundary is a jagged line [44,45,46,47]. The dynamics of this jagged (zigzag) current boundary can lead to turbulent formations along the front and circular currents (vortices). Moreover, these vortex formations can also keep the opposite direction of the circulation of currents. Therefore, the destruction of the critical state and the breaking the flux off a dome-like profile of induction can be accompanied by a dynamic “mixture” of oppositely directed electromagnetic fields, which leads to a more complex spectral composition of the induction signal than in the screening mode. This electromagnetic mixture, superimposed on the “resonator manifestations” of the plate, can strongly influence the triggering of first finger and steady state of cascades generation. As a consequence, a scatter-swing of experimental points in field dependences of the *T*_c_(*H*_ext_) periods (Figure 6b,d) and frequencies in the *F*_FFT_(*H*_ext_) spectra (Figure 8c,d) may appear.

The possible physical reasons of the complicated signal spectrum, accompanying avalanche flux dynamics, can be associated with the structure roughness of the moving flux front and inhomogeneous relief of induction. The electromagnetic radiation, accompanying the avalanche, is formed as a result of the acceleration and deceleration of individual sections of the front of moving flux, during interaction with fixed vortex bundles, followed by their separation from the pinning centers. A non-monotonic increase in the amount of flux entering the superconductor may occur and a moving flux can sweep away pinned flux hills (vortex bundles), like a tsunami, as simulation of thermomagnetic avalanches demonstrates [40]. In this case, the magnitude of the pinning inhomogeneity determines the level of electromagnetic “noise”, accompanying the flux dynamics. The magnitude of inhomogeneities can decrease, with an increase in magnetic field, due to a decrease in the critical current density. In the third quadrant, when the direction of the external field changes, regions of the superconductor arise, where the magnetic field induction has the opposite direction of the field lines compared to the trapped flux. In the fields of 0–−2T, this can be strongly reflected in the excitation conditions and, accordingly, in the spectrum of avalanche pulses. In stronger fields µ_0_*H*_ext_ ≥ −2 T, the induction of external field direction only remains in the plate and its distribution is similar to that in the first quadrant [43], leading to the coincidence of frequency dependences *F*_ma_(*H*_ext_) of cascades (Figure 9a) for two quadrants.

## 4. Conclusions

The cascades of magnetic flux avalanches at the remagnetization of superconducting bulk plates, with the magnetic flux Φ entering (screening mode) or leaving (trapping regimes) in the form of a multistage “stairway” manner with approximately equal successive portions ΔΦ, were discovered. The width of steps in temporal dependence Φ(*t*) was found to increase almost linearly with an increase of the plate thickness. Cascades in the *screening mode* after ZFC are realized in the form of fairly stable flux steps in the entire range of the instability of critical state. In this case, the number of flux steps gradually increases, with an increase in the field, reaching ten in the field range of 2–4 T. When the magnetic field is ramping backward (flux *trapping mode*) and then its orientation is changed to the opposite, the structure of the cascades becomes much more complicated.

Spectral analysis of the inductive signal allowed us to characterize the avalanche flux dynamics in the second and third quadrants in the entire region of critical state instability. It was found that, in the magnetic field range of ±2–±4 T, an avalanche is realized in a form of multistage “stairway” manner (steady state of cascades generation). In the 0–2 T region, stable cascades with small quantity of steps were rarely observed. Here, the signal spectrum consists of several spectral components with approximately equal amplitudes. Such a spectrum can arise as a result of an avalanche rearrangement of the current configuration of the critical state under the action of magnetic flux front. In the second quadrant, there are two powerful current circuits with the opposite direction of circulation. These are the shielding current and flux trapping current circuits.

The possible physical reasons of the complicated signal spectrum accompanying avalanche flux dynamics can be associated with rough structure of moving flux front and inhomogeneous relief of induction, formed as a result of previous avalanche processes. In the third quadrant, when the direction of the external field changes, the regions of superconductor arise, where the induction has the opposite direction of the field lines compared to the orientation of the trapped flux. This, in the fields of 0–−2 T, is strongly reflected in the conditions of excitation of cascades and, accordingly, in the spectrum of avalanche pulses. In stronger fields µ_0_*H*_ext_ ≥ 2 T, the induction of external field direction only remains in the plate. In this region of fields, the distributions of induction in the plate in the third and first quadrants behave in the same way. Similarity in behavior is characteristic of the periodicity of cascades.

The use of various sensors allowed us to establish that, the initiation of cascades occurs mainly in the central part of the lateral surface of the plate. The speed of propagation of an avalanche front along the magnetic field line in a superconductor was experimentally determined and possible mechanism of cascades generation, related to the “resonator’s properties” of plate, was suggested.

## Figures and Tables

**Figure 1 materials-15-02037-f001:**
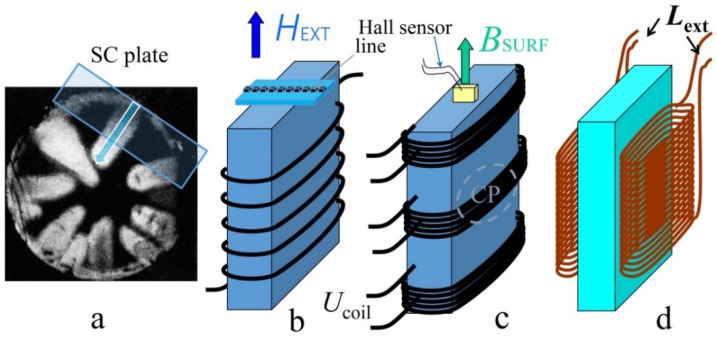
(**a**) Flux jump frozen in the form of “fingers”; Nb disc, *T* = 1.7 K (Figure 1a from Ref. [13]). The arrangement of pickup coils on superconducting plate for the avalanche flux registration: (**b**) in the whole plate’s volume, (**c**) on three height levels, and (**d**) in magnetic stray field (*L*_ext_). Hall sensor line and probe for magnetic induction (*B*_surf_) measurement are presented in (**b**,**c**); *U*_coil_ (*t*)—voltage on the pickup coil; CP is the central part, where the avalanches are mainly triggered.

**Figure 2 materials-15-02037-f002:**
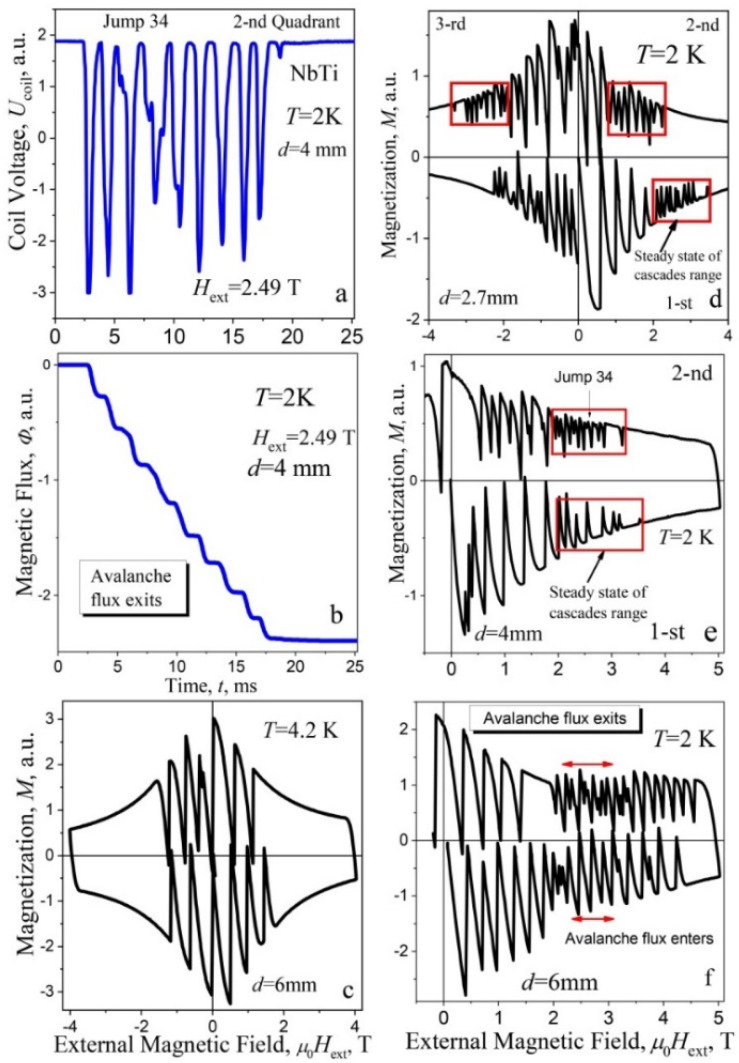
(**a**) The voltage *U*_coil_(*t*) at the flux cascade in a trapping regime [Jump 34 on hysteresis loop (**e**)]; (**b**) time-integrated voltage *U*_coil_(*t*)—a multi-step “stairway” structure of magnetic flux Φ(*t*)); (**d**–**f**) remagnetization curves *M*(*H*_ext_) for the plates with the different thickness: (**d**) *d* = 2.7 mm, (**e**) 4 mm, (**f**) 6 mm at the temperature of 2 K; the ranges of cascades are shown by the rectangles and arrows on hysteresis loops. (**c**) Hysteresis loop at 4.2 K, *d* = 6 mm.

**Figure 3 materials-15-02037-f003:**
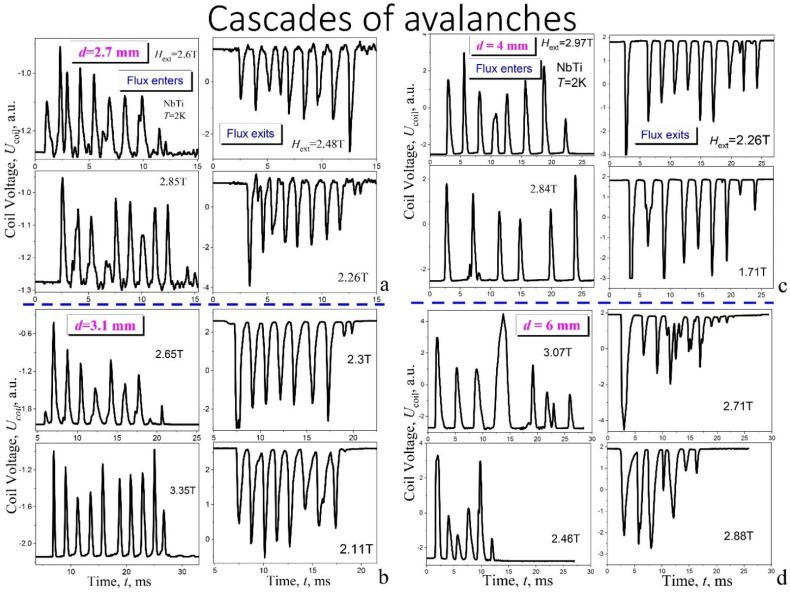
A panorama of the voltage impulses *U*_coil_(*t*) at the flux cascades for the plates with various thickness: 2.7 mm (**a**), 3.1 mm (**b**), 4 mm (**c**), and 6 mm (**d**). Each panel presents cascades recorded in four various fields and in the left column there are cascades in screening mode (1st quadrant of hysteresis loops) while in the right one—in trapped mode (2nd quadrant).

**Figure 4 materials-15-02037-f004:**
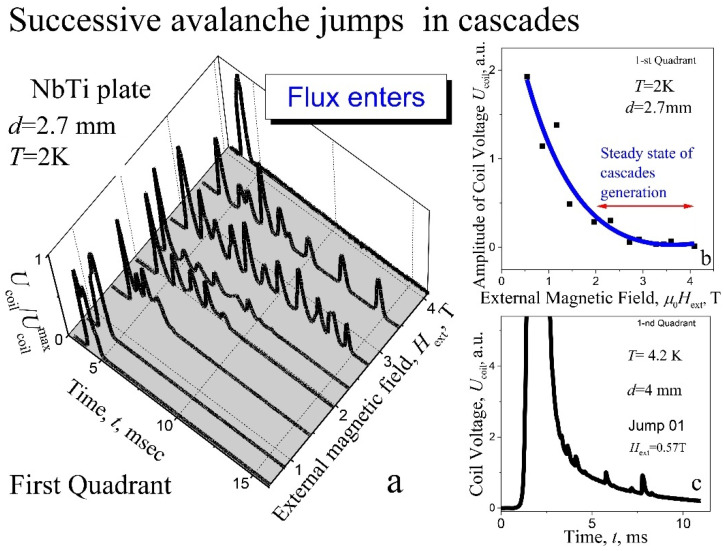
(**a**) Evolution of the normalized voltage impulses structure *U*_coil_(*t*)/*U*^max^_coil_ at the avalanche during the increase of magnetic field; (**b**) the amplitude of voltage impulse vs magnetic field and exponential fitting to the experimental data: *d* = 2.7 mm, *T* = 2 K; (**c**) the voltage impulse at first flux jump: *T* = 4.2 K, µ_0_*H*_ext_ = 0.57 T, *d* = 4 mm. Screening regime; ZFC.

**Figure 5 materials-15-02037-f005:**
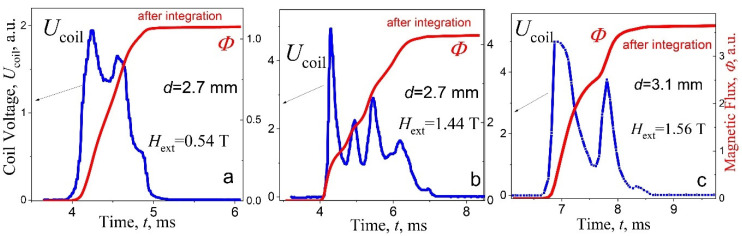
(**a**–**c**) The voltage impulses *U*_coil_(*t*) (left ordinate) and magnetic flux Φ(*t*) steps (right ordinate) at the avalanches: (**a**) first jump (µ_0_*H*_ext_ = 0.54 T), (**b**) fourth jump (µ_0_*H*_ext_ = 1.44 T) for the plate with thickness *d* = 2.7 mm; (**c**) fourth jump (µ_0_*H*_ext_ = 1.56 T), *d* = 3.1 mm; ZFC, screening regime (1st quadrant), *T* = 2 K.

**Figure 6 materials-15-02037-f006:**
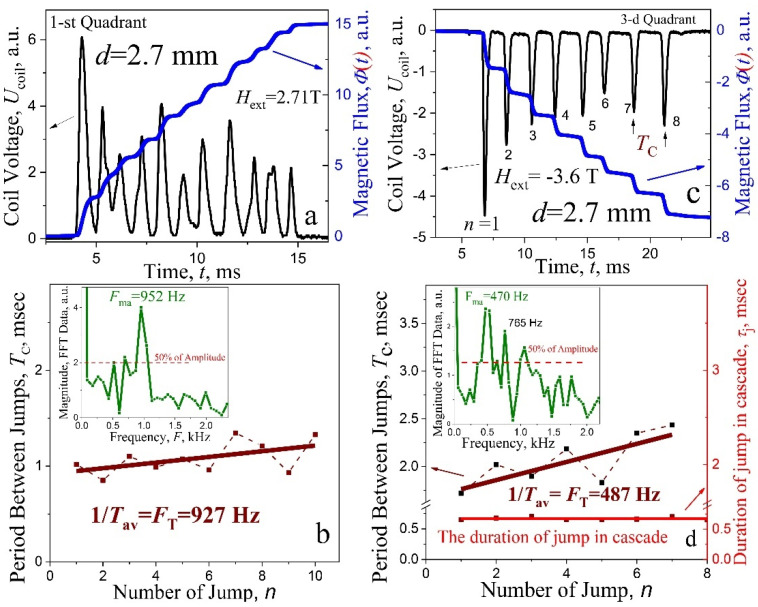
(**a**,**c**) The voltage impulse *U*_coil_(*t*) at the avalanche in screening (**a**) and in trapping (**c**) regime (left ordinate) and multi-step “stairway” structure of magnetic flux Φ(*t*) (right ordinate); (**b**,**d**) main frame—periods *T*_c_ between jumps in cascade vs number of jumps *n*. The duration of separate flux impulses in the cascade is given by bottom line in (**d**). Data for the magnitude in fast Fourier transformation of signal *U*_coil_(*t*) are presented in the inserts (**b**,**d**); *d* = 2.7 mm, *T* = 2 K.

**Figure 7 materials-15-02037-f007:**
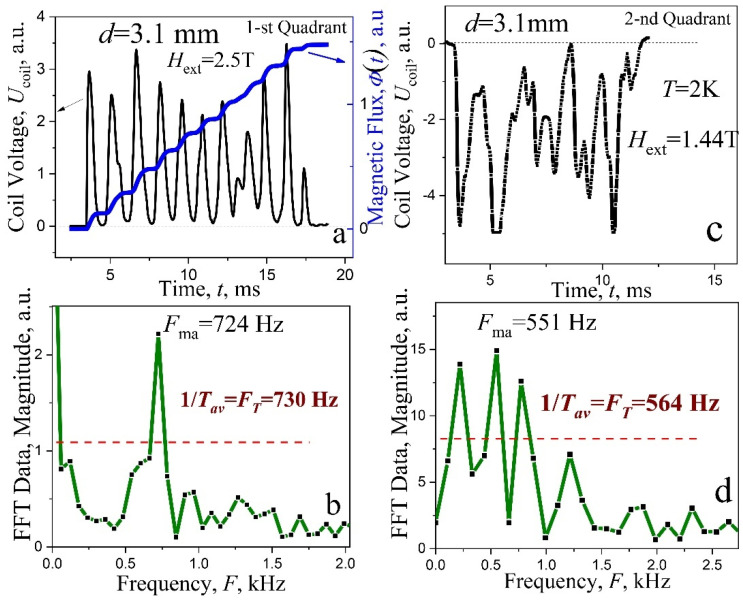
(**a**,**c**) The voltage impulse *U*_coil_(*t*) at the avalanche in trapping (**a**) and screening (**c**) regime (left ordinate) and multi-step “stairway” structure of magnetic flux Φ(*t*) [right ordinate in (**a**)]; (**b**,**d**) magnitude data from fast Fourier transformation of signal *U*_coil_(*t*); *d* = 3.1 mm, *T* = 2 K.

**Figure 8 materials-15-02037-f008:**
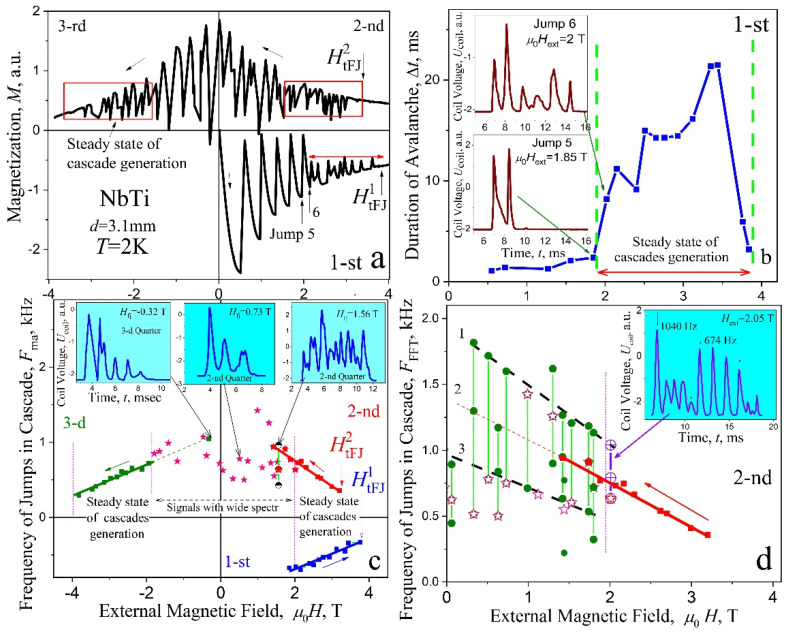
(**a**) Remagnetization curve *M*(*H*_ext_) of superconducting plate; the ranges of steady state cascades generation are shown by the rectangles and arrows. (**b**) Main frame—the duration of avalanches Δ*t*(*H*_ext_) in screening mode; insert—the structure of jump 5 and jump 6. (**c**) Main frame—frequencies *F*_ma_ with the maximal amplitude in specters of signals vs. magnetic field for different quadrants; inserts—the voltage impulses induced at avalanches in fields µ_0_*H*_ext_ = 1.56 T and 0.73 T (2nd quadrant) and −0.32 T (3rd quadrant). (**d**) Main frame: the frequencies *F*_ma_ marked by asterisk (☆) and frequencies of components with the amplitude closest in the value vs magnetic field marked by full circles (●); insert—the structure of signal in field µ_0_*H*_ext_ = 2.01 T with two characteristic pronounced frequencies, 1040 Hz and 674 Hz, marked by crossed open circles (⊕), 2nd quadrant. *d* = 3.1 mm, *T* = 2 K.

**Figure 9 materials-15-02037-f009:**
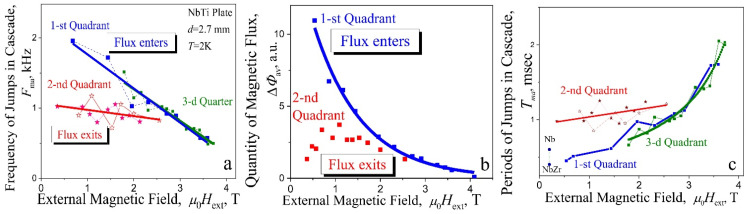
(**a**) The frequencies *F*_ma_ with the maximal amplitude in specters of signals vs magnetic field for different quadrants. (**b**) Quantity of the magnetic flux enters (1st quadrant) and exits (2nd quadrant) at avalanches vs magnetic field; (**c**) the period *T*_ma_ = 1/*F*_ma_ between jumps in cascade vs magnetic field for different quadrants; *d* = 2.7 mm, *T* = 2 K. Symbol (●)—the time during which the avalanche flux front travels a distance of 3 mm in Nb and NbZr at 1.6 K and at magnetic field µ_0_*H*_ext_ = 0.2 T [14].

**Figure 10 materials-15-02037-f010:**
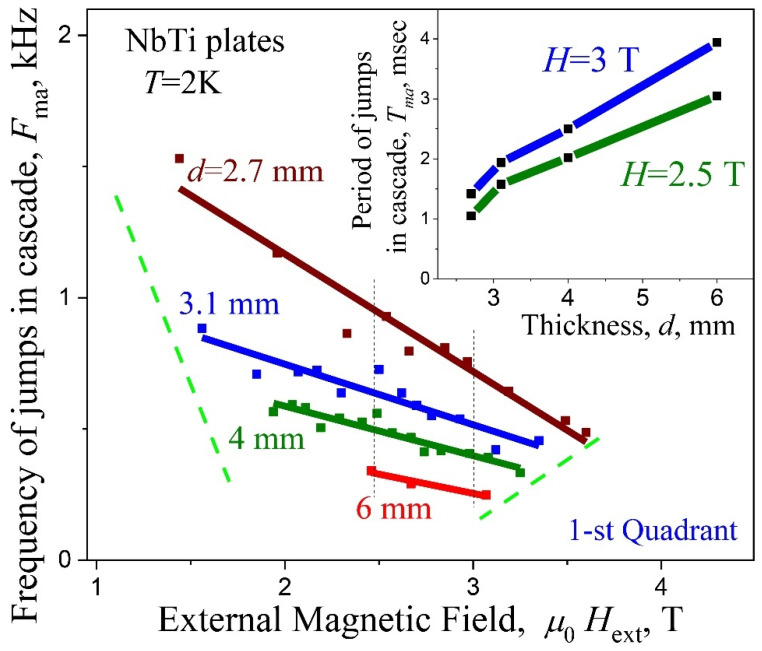
Frequency *F*_ma_ vs. *H*_ext_ for plates with different thickness *d*; screening regime (1st quadrant). Insert—the value of period *T*_ma_ = 1/*F*_ma_ along vertical lines at the magnetic field µ_0_*H*_ext_ = 2.5 and 3 T on main frame vs. thickness *d* of plate.

**Figure 11 materials-15-02037-f011:**
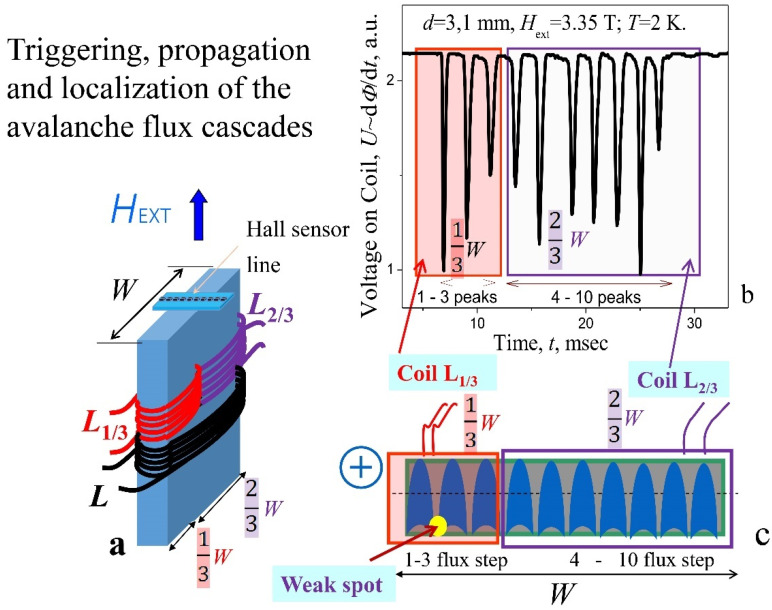
(**a**) The arrangement of pickup coils for the avalanche flux registration in the whole plate’s width (*W*) − (*L*), in the 1/3*W* (*L*_1/3_) and in 2/3*W* (*L*_2/3_); (**b**) one of the cascades recorded simultaneously by three coils *L*, *L*_1/3_, and *L*_2/3_; *d* = 3.1 mm, *H* = 3.35 T, *T* = 2 K. Full signal registered by coil *L* is divided proportionally between the coils *L*_1/3_ and *L*_2/3_, confirming spatial separation of impulses in cascades. (**c**) A schematic cross-section of a plate oriented orthogonally to magnetic field and spatial localization of successive flux avalanches. Weak spots appear in the concave corner, near the finger’s gate.

**Figure 12 materials-15-02037-f012:**
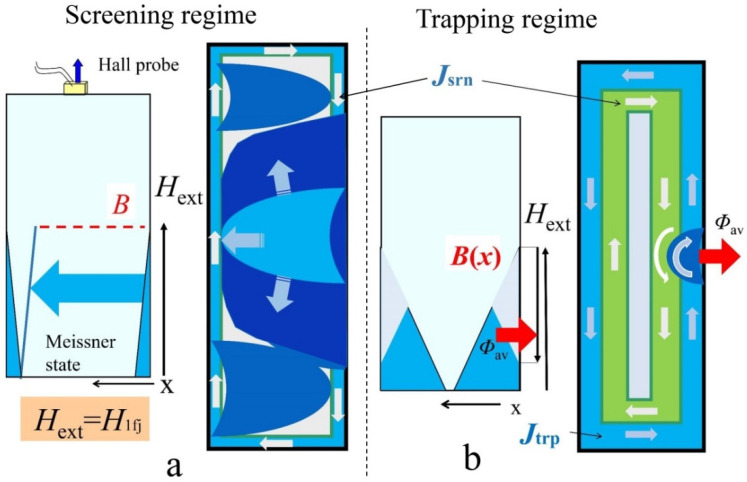
(**a**,**b**) Left side—a magnetic induction *B*(*x*) of Bean’s profile into plate before the 1st flux jumps in screening regime (**a**) and before the cascade in trapping regime appears (**b**); right side is cross section of plate perpendicular to magnetic field direction: a schematic representation of the flux avalanche entry process, corresponding to the 1st flux jump (**a**) and to the cascade creation (**b**).

## Data Availability

The data that support the findings of this study are available from the corresponding author upon reasonable request.

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
