# Peer review of "Multi-Steps Magnetic Flux Entrance/Exit at Thermomagnetic Avalanches in the Plates of Hard Superconductors"

_materials, 2022, doi:10.3390/ma15062037_

Round 1

Reviewer 1 Report

In this paper, the authors communicate an interesting discovery in the superconductors field. It is about of avalanche cascades of magnetic flux have been detected at thermomagnetic instability of the critical state in the plates of Nb-Ti alloy.

The manuscript is well documented (47 references), clear written and interesting for the researchers from this field. Results of the experiments are shown, properly analysed and explained. Conclusions are clear, substantially, in line with the main text.

However, there are two issues to be addressed:

  1. No evidence of the physical experiment is provided and should be (photos).
  2. No practical application of the discovery is shown.

Author Response

Comments and Suggestions for Authors

In this paper, the authors communicate an interesting discovery in the superconductors field. It is about of avalanche cascades of magnetic flux have been detected at thermomagnetic instability of the critical state in the plates of Nb-Ti alloy.

The manuscript is well documented (47 references), clear written and interesting for the researchers from this field. Results of the experiments are shown, properly analysed and explained. Conclusions are clear, substantially, in line with the main text.

However, there are two issues to be addressed:

  1. No evidence of the physical experiment is provided and should be (photos).

RESPONSE: The physical experiment presented in current manuscript contains two innovative parts. The first one is related to simultaneous acquisition of time dependent voltage Ucoil(t) in four channels with time resolution of ~ 10−7 s. It was reached with an industrial 4-channel NI PCI DAQ-6115 S card connected to the computer, as it is described in the manuscript. The second innovative part is related to the sensors applied in the measurements. The system of original handmade coils was prepared for each plate configuration and then it was modified/destroyed when the thickness of the plate was changed. According to us, schematic drawings of the design of such coils’ systems used in our experiments are much more informative than their photos and therefore, the drawings only were presented in the manuscript on Fig. 1. The rest of our experimental setup consisted of commercially available DAQ board, mounted in PC computer as well as of a magnet with cryostat and thus, photos of physical experimental setup were not introduced into the manuscript.

  1. No practical application of the discovery is shown.

RESPONSE: The practical implication of the discovery was already briefly described in original manuscript (page 10 – lines 299–305). However, according to the Referee remark, the following sentences: “The discovery of a multi-steps magnetic flux entrance/exit at thermomagnetic avalanches in the plates of hard superconductors could make a difference in the case of applications of electric motors or generators built with superconducting elements in the form of plates. In the plates with certain thicknesses, this phenomenon reduces the risk of failure at maximum loads.” were introduced into current version of the manuscript.

Reviewer 2 Report

Dear Authors,

The article titled "Multi-steps magnetic flux entrance/exit at thermomagnetic avalanches in the plates of hard superconductors" submitted to Materials is well written. I recommend Minor corrections of the article with Minor English Corrections.

1. Why the metals Nb-Ti are used, the carbide or graphene or carbon based materials are much more stronger.

2. Please add the list of abbreviations used throughout the manuscript, such as: jc, Φ(t), ΔΦ, Nb-Ti, YBa2Cu3O7-δI etc.

3. What are irregular jumps, kindly explain the phenomenon for better readability.

4. Are there no previous studies on the multi-step stairway structure of magnetic flux dynamic at avalanches. Please add and highlight the novelty statement correctly. 

5. If in case Figure 1 is derived from previous literature, please cite.

6. What are the main outcomes of the study and practical purpose?

7. Please add all the materials used, dimensions, content, size, equipment used in the Materials section.

8. Remove the background colour from all the figures, you can also add RSM, AI, ML for the reliability, repetitions and validation of the study.

Thank you!

Author Response

Comments and Suggestions for Authors

The article titled "Multi-steps magnetic flux entrance/exit at thermomagnetic avalanches in the plates of hard superconductors" submitted to Materials is well written. I recommend Minor corrections of the article with Minor English Corrections.

  1. Why the metals Nb-Ti are used, the carbide or graphene or carbon based materials are much more stronger.

RESPONSE: The success of Nb–Ti has been due to its combination of excellent strength and ductility with high current-carrying capacity at magnetic fields sufficient for the most of applications. Strength is not the main practical property of materials; high conductive properties are required here. Superconducting Nb-Ti is the No. 1 material in the world in the terms of wide application. Material based on carbon exhibits bad conductivity. The sentence “being the number one material in the world in the terms of wide application” was introduced into manuscript.

  1. Please add the list of abbreviations used throughout the manuscript, such as: jc, Φ(t), ΔΦ, Nb-Ti, YBa2Cu3O7-δI etc.

RESPONSE: The list of abbreviations used throughout the manuscript was added.

  1. What are irregular jumps, kindly explain the phenomenon for better readability.

RESPONSE: The term “irregular” jumps was taken from Ref. [13]. As “irregular” the authors mean the jumps leading to a complex, irregular distribution of the magnetic flux in the superconductor. An appropriate explanation was added in the revised manuscript (page 2).

  1. Are there no previous studies on the multi-step stairway structure of magnetic flux dynamic at avalanches. Please add and highlight the novelty statement correctly.

RESPONSE: The experimental observation of multi-step “stairway” structure of magnetic flux dynamic at avalanches is reported here for the first time. It was clearly stated on page 3 of the manuscript. No changes were introduced into current version of the manuscript.

  1. If in case Figure 1 is derived from previous literature, please cite.

RESPONSE: The caption of Figure 1 has been corrected: “[after 13]” has been replaced by “(Fig. 1(a) from Ref. [13])”.

  1. What are the main outcomes of the study and practical purpose?

RESPONSE: To emphasize the main outcomes of the study the following paragraph was added at the end of introduction section of revised manuscript:

“The experimental results presented here demonstrate the new dynamical properties, appearing during thermomagnetic avalanches in the plates of type II superconductors. It was found that, the magnetic flux enters conventional superconductor in screening regime and leaves in trapping regime in the form of a multistage “stairways” (cascades), with the structure dependent on the magnetic field strength and magnetic history, with approximately equal successive portions in temporal dependence, and with the width depending almost linearly on the plate thickness. The mechanism of cascades generation seems to be connected to the “resonator’s properties” of the plates.”

The practical implication of the discovery was already briefly described in original manuscript (page 10 – lines 299–305). Additionally, the discovery of a multi-steps magnetic flux entrance/exit at thermomagnetic avalanches in the plates of hard superconductors could make a difference in the case of applications of electric motors or generators built using superconducting elements made in the form of plates. In the plates with certain thicknesses, this phenomenon reduces the risk of failure at maximum loads.

The last two sentences were also added in revised manuscript (Section 3.2).

  1. Please add all the materials used, dimensions, content, size, equipment used in the Materials section.

RESPONSE: The following description:

“Hot extrusion of Nb-Ti 50% alloy was carried out according to the standard technology [37] through a deformation of matrix at a temperature of 750 ºС along the route from Ø 50 mm → Ø 15 mm (Ø – diameter of the rod) with a draw value R = Sbefore/Safter ≈ 11, where Sbefore and Safter are the areas of the sample sections before and after deformation, respectively.”,

including Ref. [37]:

[37] L. D. Cooley, P. J. Lee, and D. C. Larbalestier, Conductor processing of low-Tc materials: the alloy Nb-Ti, in Handbook of Superconducting Materials, Volume I: Superconductivity, Materials and Processes, Chapter B.3.3.2, Vol. 603, D.A. Cardwell and D. S. Ginley, eds., Institute of Physics Publishing, Bristol, 2003.

was added in the revised manuscript.

  1. Remove the background colour from all the figures, you can also add RSM, AI, ML for the reliability, repetitions and validation of the study.

RESPONSE: Background color was removed from all the figures. Using RSM, AI, ML is a very interesting suggestion. We will consider it in our further research.

Reviewer 3 Report

The following parts should be revised:

  1. introduction should not contain figure 1.

  2. The innovations and importance of this study should be clearly stated at the end of the introduction section.

  3. In section 2, figure 1b, 1c,1d should appear after the relevant text.

  4. In lines 165-165, the thickness should be written as 2.7 mm, 3.1mm, 4mm, and 6mm. Not -2.7mm, -3.1mm, -4mm, and -6mm.

  5. In line 197, what means ZFC?

  6. conclusion should not contain the number of figures (figure 9a).

  7. Line 562-line 592 should be divided into some mini paragraphs.

  8. the English of this manuscript should be improved.

Author Response

Comments and Suggestions for Authors

The following parts should be revised:

1. introduction should not contain figure 1.

RESPONSE: Figure 1 was moved from „Introduction” to Section 2 „Experiments and Materials”.

2. The innovations and importance of this study should be clearly stated at the end of the introduction section.

RESPONSE: The following paragraph was added at the end of introduction section of revised manuscript: “The experimental results presented here demonstrate the new dynamical properties, appearing during thermomagnetic avalanches in the plates of type II superconductors. It was found that, the magnetic flux enters conventional superconductor in screening regime and leaves in trapping regime in the form of a multistage “stairways” (cascades), with the structure dependent on the magnetic field strength and magnetic history, with approximately equal successive portions in temporal dependence, and with the width depending almost linearly on the plate thickness. The mechanism of cascades generation seems to be connected to the “resonator’s properties” of the plates.”

3. In section 2, figure 1b, 1c,1d should appear after the relevant text.

RESPONSE: In revised manuscript these figures appear just after relevant text in section 2.

4. In lines 165-165, the thickness should be written as 2.7 mm, 3.1mm, 4mm, and 6mm. Not -2.7mm, -3.1mm, -4mm, and -6mm.

RESPONSE: It was corrected.

5. In line 197, what means ZFC?

RESPONSE: ZFC means the “zero field cooling” mode, i.e. after cooling in zero magnetic field. In revised manuscript, we defined this abbreviation at the begin of Section 3.1.

6. conclusion should not contain the number of figures (figure 9a).

RESPONSE: The number of figure [Fig. 9(a)] was removed form Conclusions in revised manuscript.

7. Line 562-line 592 should be divided into some mini paragraphs.

RESPONSE: The large paragraph from Conclusions was divided into 3 smaller ones.

8. the English of this manuscript should be improved.

RESPONSE: We did our best to improve English.